# Traditional and New Routes of Trophoblast Invasion and Their Implications for Pregnancy Diseases

**DOI:** 10.3390/ijms21010289

**Published:** 2019-12-31

**Authors:** Berthold Huppertz

**Affiliations:** Division of Cell Biology, Histology and Embryology, Gottfried Schatz Research Center, Medical University of Graz, 8010 Graz, Austria; berthold.huppertz@medunigraz.at; Tel.: +43-316-385-71897

**Keywords:** trophoblast, invasion, placenta, uterine glands, uterine milk, intra-uterine growth restriction, pregnancy outcome

## Abstract

Historically, invasion of placental trophoblasts was thought to be extremely specific, only invading into the connective tissues of the maternal uterus and finally reaching and transforming the uterine spiral arteries. Only recently, identification of new routes of trophoblast invasion into different structures of the maternal uterus has been achieved. Thorough morphological analysis has resulted in the identification of trophoblasts invading into glands, veins, and lymph vessels of the uterine wall. These new routes pave the way for a re-evaluation of trophoblast invasion during normal placental development. Of course, such new routes of trophoblast invasion may well be altered, especially in pregnancy pathologies such as intra-uterine growth restriction, preeclampsia, early and recurrent pregnancy loss, stillbirth, and spontaneous abortion. Maybe one or more of these pregnancy pathologies show alterations in different pathways of trophoblast invasion, and, thus, etiologies may need to be redefined, and new therapies may be developed.

## 1. Introduction

Proper and strictly controlled invasion of extravillous trophoblasts is mandatory for placental development, enabling the normal growth of a fetus in the maternal uterus. The trophoblast cell line develops at the time of blastocyst formation and divides into two main cell populations, (1) the villous trophoblast with villous cytotrophoblasts and the multinucleated syncytiotrophoblast, forming the outer cover of all placental villi; and (2) the extravillous trophoblast that invades into the maternal uterine tissues, reaching down to the inner third of the myometrium.

Extravillous trophoblasts start their journey at trophoblast cell columns that develop at the tips of anchoring villi that attach to the uterine wall. Within these cell columns the trophoblast cells in direct contact to the villous basement membrane proliferate and build the source for all extravillous trophoblasts. Their daughter cells leave the cell cycle and are pushed toward maternal tissues by the proliferative pressure of the cells at the basement membrane. After a transitional phase, the daughter cells start their active migration and invade into the uterine connective tissues. This is why these cells have been termed “interstitial trophoblasts” [1].

Traditionally, the visualization of extravillous trophoblasts has been achieved by using antibodies against cytokeratin isoforms, such as cytokeratins 8 and 18 [2], and mostly cytokeratin 7 [3,4,5]. Although trophoblast staining for cytokeratin was always referred to as highly specific, especially in the placenta of a first-trimester placenta, other fetal and maternal cells display immunostaining for these cytoskeletal proteins, including epithelial cells of the embryonic amnion or epithelial cells of maternal uterine glands [6]. With the identification of the highly specific expression of the major histocompatibility complex protein HLA-G on extravillous trophoblasts [7,8,9] followed by the development of suitable antibodies that specifically bind to only this type of HLA proteins, a new era of identification of extravillous trophoblasts began [6,10].

As will be described below, alterations of trophoblast invasion have been associated with pregnancy pathologies, including preeclampsia, intra-uterine growth restriction (IUGR), spontaneous abortion, and placenta accrete/increta/percreta [11]. So far, scientists tried to link all the above pathologies with trophoblast invasion in total or invasion into uterine arteries only. In specific cases, hypotheses finding associations between pathology and trophoblast invasion were developed but failed testing due to conceptual challenges [12]. Similarly, the placental expression and release of growth factors such as sFlt-1 and/or PlGF were associated with specific pregnancy pathologies, but also, here, the direct link between placental growth factor expression and pathology development could not be established [12,13].

So far, the new routes of trophoblast invasion have not been investigated regarding their impact on placental and thus fetal development. Hence, there is a great knowledge gap that needs to be filled.

## 2. Historical Thinking of Trophoblast Invasion

Early descriptions of uterine spiral arteries during pregnancy date back to 1774, when William Hunter described “convoluted arteries that passed between the womb and the placenta” [14]. At that time, no one thought of placental cells invading into the maternal uterus. About a century later, Friedländer (1870) was the first to describe “endovascular cells” in these spiral arteries, without mentioning any information on the source of these cells. This observation of Friedländer was described in a book that was published another 50 years later, by Grosser (1927) [15]. This author was the one who first imagined that the endovascular cells in spiral arteries during pregnancy are not necessarily derived from the maternal decidua but rather could well be of trophoblastic/placental/fetal origin.

It was not until the 1950s and 1960s that the aspect of arterial transformation was revisited, and major observations on spiral artery transformation were published by Harris and Ramsey [16,17], as well as Boyd and Hamilton [18,19]. Both groups described perivascular trophoblast in the decidual stroma surrounding arteries, mural trophoblasts in the walls of these arteries, and intraluminal trophoblasts residing in the vessel lumen. Already at this time, it was speculated that the trophoblast cells within the lumen of spiral arteries may well be washed out into the intervillous space of the placenta.

Interestingly, both groups missed any other route of trophoblast invasion into other luminal structures of the decidua. One of the reasons for this may be the fact that, at that time, an identification of cells by using specific probes, e.g., antibodies, was not possible. Moreover, only few specimens were available at the time; and hence, knowledge on changes of spiral arteries until delivery, even during normal pregnancy, was very sparse.

This became obvious when the first studies dealt with alterations of trophoblast invasion into arteries in pregnancy pathologies. In one of these early studies the authors stated the following: “The examination of the spiral arteries in pregnancy associated with hypertension has not been easy because of two factors. The first was the difficulty of obtaining suitable material and the second the occurrence in the same spiral arteries of extensive morphological changes due to pregnancy itself” [20]. Thus, at the time when scientists and clinicians became aware of structural alterations of invaded spiral arteries in pathological cases, they realized that there was not enough knowledge on how a normal placental bed with invaded structures looks like.

The combination of the following facts may be the reason why, even today, the knowledge on trophoblast invasion is very restricted:

(1) In the very beginning, only very few groups looked into changes of uterine vessels, in particular focusing on spiral arteries.

(2) These groups visualized arteries and the entrance of blood into the intervillous space of the placenta by dye injection and thus missed the veins.

(3) These groups visualized arteries and the infiltrating trophoblasts mostly in monkeys and only a few human cases, and thus may have missed the differences between monkeys and humans.

(4) These groups had no tools in hands to specifically identify invading trophoblasts.

(5) Maybe due to the combination of the above facts, trophoblast invasion into other luminal structures never came into focus.

(6) Scientists following these initial studies simply used this knowledge as a basis and did not scrutinize the real variety of structures invaded by extravillous trophoblast.

## 3. Looking into Invaded Uterine Structures from the Embryo’s Nutritional Point of View

Looking from the side of the embryo in terms of nutritional support from the mother, shortly after implantation, there is the need to increase nutritional support due to the massively increasing volumes of embryo and placenta in the absence of any supporting blood vessel. Within the endometrium of the human uterus, this is best performed by eroding uterine glands in direct vicinity to the placenta, allowing direct contact of the syncytiotrophoblast to the glandular secretion products. Hence, it looks as if histiotrophic nutrition of the embryo already starts a few days after implantation [21].

In the collection of images of Allen Enders at the Centre for Trophoblast Research in Cambridge, there are images from case 8020, which is considered the earliest specimen in the Carnegie collection, probably of day one after initiation of implantation. In one of the images from case 8020, the margin of the trophoblastic plate is displayed (Figure 1A). Here, the initially invading syncytiotrophoblast has already invaded uterine glands underlying the embryo. This is the earliest description of glandular invasion by trophoblast. In a later-stage case of the early lacunar stage (Figure 1B), invasion into a uterine gland can be seen again [22].

Following this early invasion by the initially invasive syncytiotrophoblast, the extravillous trophoblast population takes over and further invades into uterine glands, resulting in opening these luminal structures toward the developing intervillous space of the placenta [21]. As soon as the intervillous space of the placenta is established, the glandular secretion products flow into this space and are transferred from the placenta to the embryo [23]. At the same time, the remaining secretion products and the respective fluids need to be drained back into the maternal system. Hence, erosion and connection of uterine veins to the intervillous space of the placenta needs to take place next (Figure 2A) [24,25,26]. Other images of the Enders collection show the junctional zone of trophoblast invasion at the secondary villus stage. Here, invasion into veins and glands can be found, while arteries next to these two luminal structures do not show any signs of invasion [22].

Finally, around mid-first trimester spiral arteries are the next target of the invading extravillous trophoblasts. While glands and veins have thin walls and only need to be eroded and connected to the placenta (Figure 2A), the arterial walls need to be prepared prior to invasion into them. Finally, the spiral arteries are invaded as well, and their lumen is plugged until the beginning of the second trimester (Figure 2B) [27,28].

Hence, during the first trimester, a plasma flow from the plugged arteries, plus a flow of glandular secretion products, enters the intervillous space of the placenta, which is drained back into the maternal system by the utero-placental veins (Figure 2B). This allows the nutritional support of the embryo during the first trimester of pregnancy with substances in maternal plasma, plus the secretion products of the uterine glands. This has been termed histiotrophic nutrition by glands rather than vessels [23].

While the histiotrophic nutrition seems to be sufficient in the first trimester of pregnancy, with the massive growth of the fetus later in pregnancy, a different nutritional support is needed. With the dissolution of the plugs from the arteries and the establishment of the flow of maternal blood into the placenta at the beginning of the second trimester, the nutritional supply of the fetus changes from a histiotrophic to a hemotrophic nutrition (Figure 2C) [23,29]. At the same time, the number and the input of uterine glands diminish, while, of course, the veins remain, to drain back maternal blood into the maternal circulation. Due to the lack of normal placental-bed specimens of the time around mid-gestation, it is not clear when the glandular connection to the placenta disappears. It seems as if this occurs around week 20 of pregnancy, but this still needs further elucidation.

## 4. New Routes of Trophoblast Invasion

All the above considerations are only conceivable due to the recent progress in the identification of new routes of invasion of extravillous trophoblasts (Figure 2) [30]. Interestingly, the identification of these new types of cellular pathways can only be performed by using the original tissue organization. Any dissolution of the tissue would have destroyed the possibility to identify these routes.

Of course, this is a purely descriptive approach, which needs to be followed by the elucidation of the functional differences of the cells in the different routes and pathways. It needs to be clarified whether the cells already “know” from the beginning where to go and which luminal structure they will go for or whether they just invade uterine tissues and reach a luminal structure simply by chance.

### 4.1. Endoglandular Trophoblast

The aspect of histiotrophic nutrition raised the question of how glands should release their secretion products into the placenta when there is no connection between glands and placenta [23]. Also, histiotrophic nutrition of the embryo/fetus by secretion products of uterine glands was referred to as “uterine milk” in those eutherians with an epitheliochorial placentation, including animals such as the sheep, cow, and pig [31]. So far, histiotrophic nutrition has not been described in eutherians with a hemochorial placentation, such as in humans, rats, and mice.

The identification of a subpopulation of extravillous trophoblasts invading into uterine glands allowed the aspect of histiotrophic nutrition to come into focus also in the human [5,32]. The new subpopulation of endoglandular trophoblasts does not only invade into uterine glands but also connects them to the placenta (Figure 2A) [33], thus allowing early nutritive support of the embryo by using the secretion products of the uterine glands. This is the first description of the “uterine milk” in a eutherian with hemochorial placentation, the human.

As outlined above, invasion into uterine glands already starts prior to the establishment of the extravillous trophoblast cell population. Images from very early time points of human implantation depict invasion of the early invasive syncytiotrophoblast into uterine glands as early as one day after implantation [21,22,32]. Hence, it seems important for the embryo to have this histiotrophic supply right from the beginning of pregnancy, before other means to nutritive support take over at the end of the first trimester.

Another aspect of endoglandular invasion is the interesting observation of the escape of these trophoblast cells from the placental bed. As outlined above, during the first half of pregnancy, endoglandular trophoblasts invade into uterine glands. This also takes place at the outer margin of the growing placenta. At this site, some of the uterine glands may already have been invaded and connected to the intervillous space of the placenta, while other glands are still open to their normal target, the uterine cavity. If endoglandular invasion takes place into glands that are already connected to the intervillous space of the placenta, then trophoblast cells that are washed out from the glands enter into the intervillous space and are then drained into the vascular system of the mother. However, if glands are invaded that are still connected to the uterine cavity, endoglandular trophoblasts that invade into such glands and are washed out may end up being flushed into the uterine cavity. From here, they can easily reach the cervix, from where they can be isolated and used for noninvasive prenatal testing [34]. Interestingly, the cervix during the first half of pregnancy seems to be the site with the highest density of extravillous trophoblasts, and hence it is tempting to use these cells for noninvasive prenatal testing [33].

### 4.2. Endovenous Trophoblast

Following invasion into uterine glands, it seems as if the next route of invasion guides extravillous trophoblasts toward uterine veins (Figure 2A) [24,25,26]. The secretion products of the uterine glands need to be removed from the growing intervillous space. With the invasion of uterine veins by endovenous trophoblasts and the connection of the veins to the placenta, this removal is assured (Figure 2A). The connection of the veins prior to the connection of the arteries is physiologically allegeable as removal of the blood plasma flowing into the placenta needs to be secured prior to floating of the intervillous space (Figure 2B). As outlined above, respective findings have been obtained already during very early stages of placentation [22].

### 4.3. Endoarterial Trophoblast

The next route of invasion is the one that has been identified centuries ago, invasion into spiral arteries by endoarterial trophoblasts (Figure 2B) [21]. Here, invasion is much more complex than that into glands and veins. In the latter two, invasion simply needs to connect these thin-walled luminal structures to the placenta. In case of the arteries, the walls of these arteries need to be restructured and invasion goes much deeper than in veins and glands—as far as we know today (Figure 2C). There is no need for the invading trophoblasts to restructure the walls of glands and veins; there is only the need to open and connect these luminal structures to the intervillous space of the placenta. However, this is different for uterine arteries, where the muscle layer of their walls needs to be restructured and the arteries need to become large tubes that have lost their contractile abilities.

### 4.4. Endolymphatic Trophoblast

Finally, invasion into uterine lymph vessels is described (endolymphatic trophoblast) [24,26]. The function of this route of invasion is unclear so far. It may simply show that trophoblast invasion is not specific at all, and, thus, extravillous trophoblasts simply invade all luminal structures within the placental bed. However, it may well serve a function such as connecting lymph vessels to the placenta, as well to serve as additional regulatory structure to adapt intra-placental fluid pressure. In both cases, it will be interesting to see whether endolymphatic trophoblasts can be retrieved from local lymph nodes. The first data showing a respective localization have already been published [26].

## 5. Alterations of Trophoblast Invasion and the Putative Effects on Pregnancy Outcome

First insight into alterations of the migratory routes of extravillous trophoblast in pathological pregnancies is slowly evolving. Since trophoblast invasion and its alterations have only been recognized in arteries so far [34], there is only very little data available on how altered invasion into other luminal structures of the placental bed may affect pregnancy outcomes. Of course, it is easily comprehensible that failure in connecting uterine veins to the placenta leads to spontaneous abortion of the embryo early in gestation. However, the complex interplay between the different luminal structures and their invasion opens a much broader field to finally understand the effects of altered trophoblast invasion.

### 5.1. One Example of Non-Arterial Changes of Trophoblast Invasion in a Pregnancy Pathology

So far, there is only one example available, which is based on new data in the field of recurrent spontaneous abortion. In this pregnancy pathology, alterations of trophoblast invasion have been shown to be related to vascular changes. In cases with idiopathic recurrent spontaneous abortion, a role for alterations of trophoblast invasion into spiral arteries has been described; however, this role is still debated today, with no final conclusion whether or not there is a direct relation between altered arterial invasion and the etiology of recurrent spontaneous abortion [35,36,37].

Windsperger et al. (2017) [26] recently analyzed decidual placental bed tissues from cases with recurrent spontaneous abortion. These authors quantified the spatial distribution of extravillous trophoblasts in placental bed spiral arteries, veins, and lymph vessels. They identified alterations in vascular invasion only in veins and lymph vessels, hence, in non-arterial vessels [26], while invasion into spiral arteries was not affected. In cases with recurrent spontaneous abortion, there were fewer invaded lymph vessels and veins compared to the total number of such vessels in healthy controls [26].

As for all such cases with alterations of trophoblast invasion, it still needs to be clarified whether the defect is directly related to a respectively dysregulated trophoblast phenotype or whether the dysregulation is found in the uterine (micro-) environment. The study above also revealed that the decidual tissues of cases with recurrent spontaneous abortion comprise a significantly higher number of all types of vessels compared to gestational-age-matched controls [26]. This is in line with data from Quenby et al. (2009) [38], who showed an enhanced density of blood vessels in the nonpregnant secretory endometrium of women diagnosed with recurrent spontaneous abortion. Thus, more thorough and specific analyses of vessel types and subtypes of extravillous trophoblast need to be performed to decipher the still blurry picture of trophoblast invasion in pregnancy pathologies, such as recurrent spontaneous abortion.

### 5.2. General Considerations of Changes of Trophoblast Invasion and Their Effects on Pregnancy Outcome

Other examples of non-arterial changes of trophoblast invasion in pregnancy pathologies have not yet been published, as the identification of the new routes of trophoblast invasion with all its aspects has only recently been published. At the same time, the new routes of invasion open new avenues to decipher if pregnancy pathologies, such as intra-uterine growth restriction (IUGR), preeclampsia, early or recurrent pregnancy loss, stillbirth, and spontaneous abortion, may at least be partly related to abnormal trophoblast invasion into one or more uterine luminal structures. Table 1 gives an overview of which invasion failure may be related to what type of pregnancy pathology. Of course, biology always goes the most complex way; hence, it may be the balance between, e.g., invaded arteries versus invaded veins, that makes the pathology rather than the simple total number of invaded vessels per vessel type. To make the story even more complex, there is much more to look at that needs to be taken into account, including the depth of invasion in arteries, the number of connected (not only invaded) luminal structures, and the development of invasion during the whole duration of pregnancy.

## 6. New Omics Technologies and Morphological Assessment of Tissues

The recent development of new omics technologies has revolutionized our understanding of different cell types within a tissue. This is especially true for the RNA level, including technologies such as single-nucleus RNA sequencing per droplet (DroNc-Seq) [39] or single-cell combinatorial indexing RNA sequencing (sci-RNA-seq) [40]. Over the last few years, the respective technologies have been introduced to and have been used in the placenta field as well. Surveys on the cellular composition of the first-trimester placenta and decidua have now added new information on the different cell types within these tissues [41,42,43].

At the same time, the preparation of the single-cell suspensions needed for RNA sequencing technologies includes the dissociation of tissues to allow single-cell RNA sequencing. It needs to be stressed at this point that this dissociation step hinders the visualization of the single-cell microenvironment and thus the identification of the direct cell–cell interactome. In the survey publications, e.g., [41,42,43], cells are grouped based on the similarities in their RNA expression profiles. Hence, in vivo tissue neighborhoods, the original microenvironment and the direct cell–cell interactome can no longer be identified and taken into consideration. Especially in such a complex organ as the placenta, cells with a similar RNA expression profile may localize at different sites within the organ.

To identify the direct cell–cell interactome, the classical morphological analysis with immunohistochemistry for proteins or techniques such as the in situ padlock method for RNA [44,45] need to be performed. A first publication based on the use of in situ padlock probes to visualize the distribution of single mRNA species in cells still residing within their original tissues was recently published [46]. Only the combination of the RNA profile of single cells, plus their morphological mapping, will allow the correct interpretation of the cellular interactomes.

Moreover, even today, the routine morphological analysis of a tissue is performed on a section of the tissue, i.e., in only two dimensions. However, the information on the third dimension is of course crucial to fully understand the structural and thus functional interactions of cells and their surrounding matrices within a tissue. The field of 3D analysis of placental tissues is just starting to emerge, and it will take some time until the techniques used in this field can be applied to reach quantitative results. An example was recently published by Perazollo et al. (2017) [47].

Hence, even in the times of all the new omics technologies, a direct correlation of single-cell RNA profiles and the exact morphological localization of a cell is yet to be established.

## 7. Conclusions

New morphological data identified new routes of trophoblast invasion, and, thus, there is room to speculate over new and different subtypes of extravillous trophoblast. So far, it is not clear whether the extravillous trophoblast simply invades all luminal structures of the placental bed using a single phenotype, or whether there are specific trophoblast phenotypes invading arteries, veins, glands, and lymph vessels.

As the new routes of trophoblast invasion have only discovered very recently, information on effect of these routes on normal placentation and, thus, fetal development is scarce. The next years need to show how altered invasion into the different types of uterine structures may affect pregnancy outcome. There may be a large variety of pregnancy pathologies that is directly related to alterations of trophoblast invasion in arteries, veins, glands, or lymph vessels. This may not only increase our knowledge on basic processes of human development; it may also result in new therapeutic interventions based on this knowledge.

## Figures and Tables

**Figure 1 ijms-21-00289-f001:**
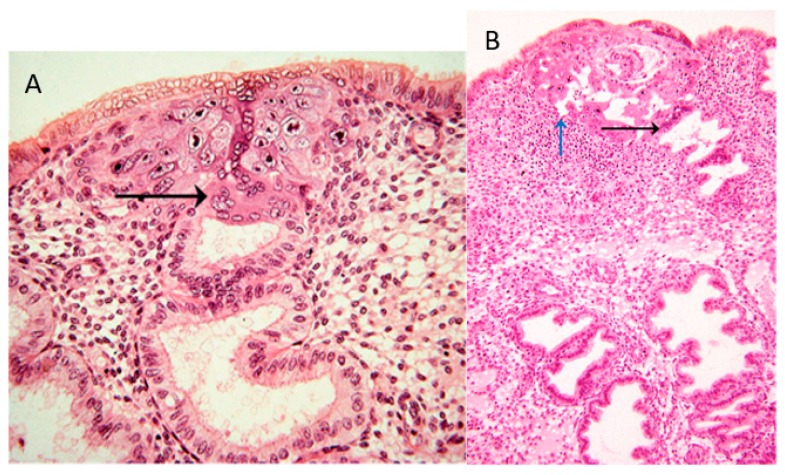
(**A**) Image #7 of case 8020: Margin of the trophoblastic plate. Allen Enders explained: “Syncytial trophoblast with small nuclei has invaded the underlying endometrial gland. It is not known whether the small nuclear syncytium is synctiotrophoblast or is partially a heterokaryon involving fusion of trophoblast and uterine cells.” The black arrow points to invasion into a uterine gland. (**B**) Image #13 of case 8171: Early lacunar stage (stage 5B). Allen Enders’ explained: “Note that the appearance of endometrial glands is similar to that seen in one of the stage 5A sites.” He further explained (under image #14 of case 8171): “Note continuity of a capillary with a lacuna that anastomoses with other lacunae. Trophoblast appears to be invading a gland in the upper right.” The black arrow points to invasion into a uterine gland, while the blue arrow points to invasion into a uterine blood vessel. Image are provided by courtesy of Allen C. Enders and the Carnegie Collection.

**Figure 2 ijms-21-00289-f002:**
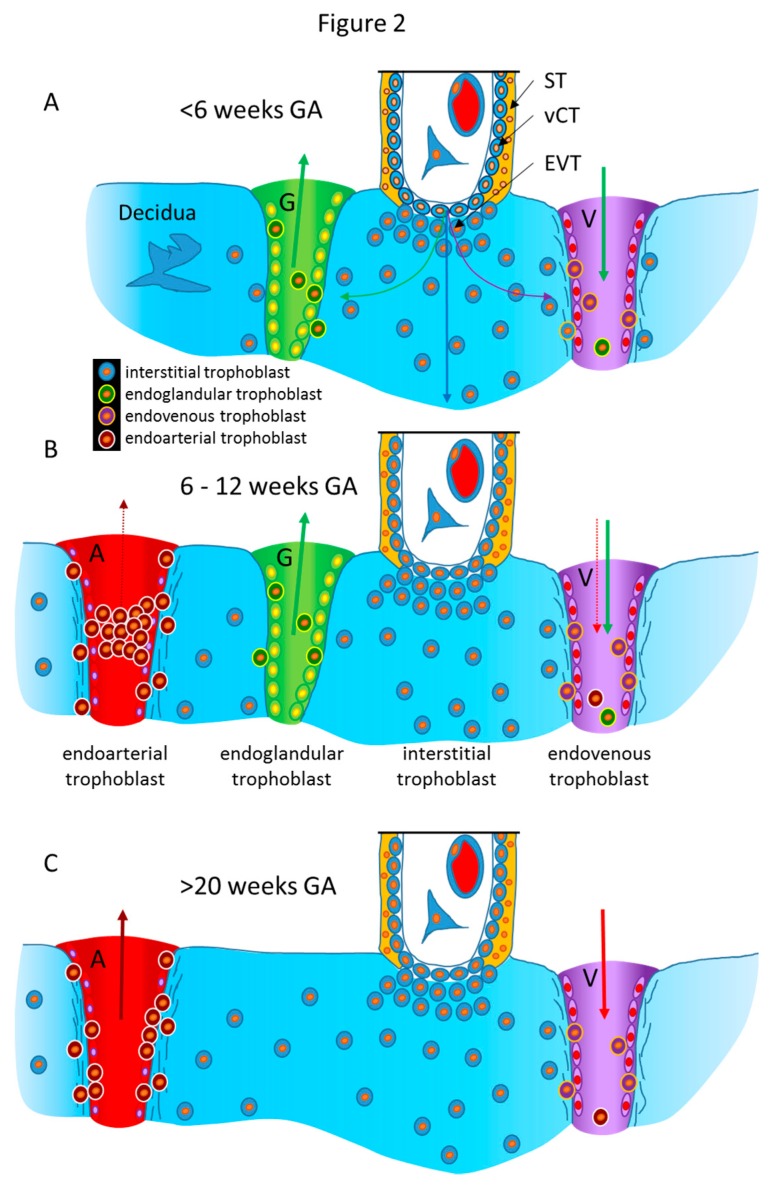
Schematic representation of the routes of trophoblast invasion during normal pregnancy. (**A**) Very early in pregnancy, prior to six weeks of gestation, invasion of the early invading syncytiotrophoblast during implantation, as well as invasion of early extravillous trophoblasts, results in opening uterine glands and veins toward the intervillous space of the placenta. Endoglandular trophoblasts open uterine glands, to enable the flow of “uterine milk” toward the placenta. This is followed by invasion of endovenous trophoblasts into uterine veins, to enable backflow of fluids into the maternal system, including villous material and endoglandular trophoblasts (shown in vein). The arrows in gland and vein represent the material transported in these structures (green arrow: glandular secretion products). (**B**) Later, during the first trimester, endoarterial trophoblasts invade into uterine spiral arteries, transform their walls, and plug their lumen, to hinder flow of maternal blood into the placenta. At that stage, only blood plasma is seeping through the plugs (indicated by the dashed red arrow). During this stage of pregnancy, the backflow via utero–placental veins comprises glandular secretion products, plus plasma from the spiral arteries (green arrow plus dashed red arrow), including villous material plus endoglandular and endoarterial trophoblasts (shown in vein). (**C**) At the beginning of the second trimester, the arterial plugs disintegrate, and the flow of maternal blood into the placenta is finally established. So far, it is not clear at which time point the glandular input diminishes and disappears, but in the second half of pregnancy, respective glands can hardly be found. Hence, this schematic drawing only shows arteries and veins (red arrows: maternal blood). Now, the venous backflow contains villous material, as well as endoarterial trophoblasts (shown in vein). A, artery; G, gland; V, vein; GA, gestational age; ST, syncytiotrophoblast; vCT, villous cytotrophoblast; EVT, extravillous trophoblast.

**Table 1 ijms-21-00289-t001:** Simplified representation of the putative effects of dysregulated trophoblast invasion for the different subtypes of extravillous trophoblast.

Extravillous Trophoblast Subtype	Invaded Structure	Putative Alteration	Putative Effect	Possibly Involved Pathologies
Interstitial trophoblast	Uterine tissues (decidua & myometrium)	Reduced	Less cells invading the uterus in general	IUGR w and w/o preeclampsia
		Enhanced	Deeper invasion than normal	Placenta accreta/increta/percreta OR Maternal anemia, pregnancy at high altitude
Endoarterial trophoblast	Uterine spiral arteries	Reduced	Faster blood flow into the placenta	IUGR w and w/o preeclampsia
		Enhanced	Further widening of the arteries	Maternal anemia, pregnancy at high altitude
Endovenous trophoblast	Uterine veins	Reduced	Decreased backflow of maternal blood into the maternal system	Early pregnancy loss, IUGR, spontaneous abortion, stillbirth
		Enhanced	Increased backflow of blood into the maternal system	Mild IUGR
Endoglandular trophoblast	Uterine glands	Reduced	Decreased nutrition of the embryo	Early pregnancy loss, spontaneous abortion
		Enhanced	Increased nutrition of the embryo	LGA
Endolymphatic trophoblast	Uterine lymph vessels	Reduced	Decreased regulation of placental fluid pressure	Spontaneous abortion
		Enhanced	?	?

?, not known so far.

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
