# Peer review of "Traditional and New Routes of Trophoblast Invasion and Their Implications for Pregnancy Diseases"

_ijms, 2019, doi:10.3390/ijms21010289_

Round 1
Reviewer 1 Report
Comments to the authors
I read with interest this narrative review on new implication of trophoblast invasion (types of invasion: endoglandular, endovenous, endoarterial and endolymphatic) in the pathogenesis of pregnancy diseases. The topic is attractive, the review is of interest for the reader. The language in understandable. The figures are well presented. The tile does not fit the purpose of this narrative review.
I have minor specific comments:
-I would change the title in: Traditional and new routes of trophoblast invasion and implication in pregnancy diseases (: a narrative review)
-the abstract and in primis the introduction should be more focalized on pregnancy diseases, and then, in the current positive impact of omics technologies.
-the appearance of table 1 should be improved following journal style
-the authors may briefly analyze the relation between cytokines and growh factor release (i.e. sflt-1 and VEGF) and trophoblast invasion.
Author Response
Manuscript ID: ijms-675333
Title: Trophoblast invasion: New implications for pregnancy pathologies
Author: Berthold Huppertz
Point-to-point reply to reviewers
Reviewer #1
I read with interest this narrative review on new implication of trophoblast invasion (types of invasion: endoglandular, endovenous, endoarterial and endolymphatic) in the pathogenesis of pregnancy diseases. The topic is attractive, the review is of interest for the reader. The language in understandable. The figures are well presented. The tile does not fit the purpose of this narrative review.
Thanks!
I have minor specific comments:
-I would change the title in: Traditional and new routes of trophoblast invasion and implication in pregnancy diseases (: a narrative review)
Following the advice of the reviewer I have modified the title to: “Traditional and new routes of trophoblast invasion and their implications for pregnancy diseases“.
-the abstract and in primis the introduction should be more focalized on pregnancy diseases, and then, in the current positive impact of omics technologies.
I have adapted the abstract to better cover the content of the manuscript text. The introduction has been rewritten completely to better focus on trophoblast invasion and pregnancy pathologies. The text on the omics technologies has been shifted to the end of the manuscript.
-the appearance of table 1 should be improved following journal style
The appearance of table 1 has been improved.
-the authors may briefly analyze the relation between cytokines and growth factor release (i.e. sflt-1 and VEGF) and trophoblast invasion.
I have recently published reviews on the topic (see e.g.: Huppertz, Expert Rev Mol Diagn 2018 and Huppertz, Curr Pharm Biotechnol 2018). Hence, I have added a paragraph on the topic at the end of the introduction but have not further dealt with it as this has already been described elsewhere.
Reviewer 2 Report
This is a Review conducted by Medical University of Graz in Austria.The aim of this study is the identification of the new theories and studies about placental trophoblast cells invasion routes into different structures of the maternal uterus.
The identification of trophoblasts invading into glands, arteries, veins and lymph vessels of the uterine wall pave the way for a re-evaluation of trophoblast invasion, especially focusing on pregnancy pathologies.
Probably, the authors should describe more in detail the pregnancy outcome of trophoblast invasion as intra-uterine growth restriction, preeclampsia, early and recurrent pregnancy loss, stillbirth and spontaneous abortion.
In this review “Material and methods” is not present: the search strategy, the database, the search term for relevant publicationsand the period of the search are not specified. Please add it on.
Specific comments:
TITLE: Core point of the title is not exhaustively argued in the article. Please modify
ABSTRACT: the abstract does not clearly explains the aim of review because not respects all contents analyzed in the text
KEYWORDS: I suggest to add: pregnancy outcome; RNA
INTRODUCTION: Authors do not well explain the importance of role of RNA of trophoblast invasion and correlation with pregnancy pathologies
2.HISTORICAL THINKING OF TROPHOBLAST INVASION: extensive description the state of art about thetrophoblast invasion in literature
3. LOOKING INTO INVADED UTERINE STRUCTURES FROM THE EMBRYO’S NUTRITIONAL POINT OF VIEW: why were the images #7 and #49 f images of Allen Enders collection not added to the article? they could be interesting!
4. NEW ROUTES OF TROPHOBLAST INVASION: this discussion clearly and synthetically explains the pathological trophoblast invasion. The figure is very good and the description is accurate
5. ALTERATIONS OF TROPHOBLAST INVASION AND THE PUTATIVE EFFECTS ON PREGNANCY OUTCOME: this chapter is very interesting because it is in line with subject of study even if it focuses too much on the effects on recurrent spontaneous abortion. I suggest to also describe as invasion failure may be related to other pregnancy pathology. The table is very detailed.
CONCLUSION: The conclusions does not clearly and synthetically explains the study design and do not explain the importance of new findings on pregnancy outcome
Author Response
Manuscript ID: ijms-675333
Title: Trophoblast invasion: New implications for pregnancy pathologies
Author: Berthold Huppertz
Point-to-point reply to reviewers
Reviewer #2
This is a Review conducted by Medical University of Graz in Austria. The aim of this study is the identification of the new theories and studies about placental trophoblast cells invasion routes into different structures of the maternal uterus.
The identification of trophoblasts invading into glands, arteries, veins and lymph vessels of the uterine wall pave the way for a re-evaluation of trophoblast invasion, especially focusing on pregnancy pathologies.
Probably, the authors should describe more in detail the pregnancy outcome of trophoblast invasion as intra-uterine growth restriction, preeclampsia, early and recurrent pregnancy loss, stillbirth and spontaneous abortion.
The introduction has been rewritten completely, taking into account those pregnancy pathologies associated with abnormal trophoblast invasion.
In this review “Material and methods” is not present: the search strategy, the database, the search term for relevant publications and the period of the search are not specified. Please add it on.
As this is a narrative and not a systematic review, the respective “material and methods” section would not make sense. This is why it was not added to the text.
Specific comments:
TITLE: Core point of the title is not exhaustively argued in the article. Please modify
Following the advice of both reviewers I have modified the title to: “Traditional and new routes of trophoblast invasion and their implications for pregnancy diseases“.
ABSTRACT: the abstract does not clearly explains the aim of review because not respects all contents analyzed in the text
I have adapted the abstract to better cover the content of the manuscript text.
KEYWORDS: I suggest to add: pregnancy outcome; RNA
I have added “pregnancy outcome” to the list of keywords. As RNA is not a major focus of this review and as it is no longer present in the introduction, I have not added it as well.
INTRODUCTION: Authors do not well explain the importance of role of RNA of trophoblast invasion and correlation with pregnancy pathologies
The introduction has been rewritten completely, now focusing on trophoblast development and invasion and pregnancy pathologies. Hence, RNA is no longer present in this section.
2.HISTORICAL THINKING OF TROPHOBLAST INVASION: extensive description the state of art about the trophoblast invasion in literature
Thanks!
LOOKING INTO INVADED UTERINE STRUCTURES FROM THE EMBRYO’S NUTRITIONAL POINT OF VIEW: why were the images #7 and #49 f images of Allen Enders collection not added to the article? they could be interesting!
As suggested I have now added the two images from Allen Enders to the manuscript as figure 1.
NEW ROUTES OF TROPHOBLAST INVASION: this discussion clearly and synthetically explains the pathological trophoblast invasion. The figure is very good and the description is accurate
Thanks!
ALTERATIONS OF TROPHOBLAST INVASION AND THE PUTATIVE EFFECTS ON PREGNANCY OUTCOME: this chapter is very interesting because it is in line with subject of study even if it focuses too much on the effects on recurrent spontaneous abortion. I suggest to also describe as invasion failure may be related to other pregnancy pathology. The table is very detailed.
In the text only the alterations in recurrent spontaneous abortion (RSA) have been described as these are the only ones that are available so far. This is why the table was added to add putative effects in other pregnancy pathologies. I have also clarified that RSA is the only pregnancy pathology where alterations of the new routes of trophoblast invasion have been detected so far.
CONCLUSION: The conclusions does not clearly and synthetically explains the study design and do not explain the importance of new findings on pregnancy outcome.
The second part of the conclusions has been rewritten following the advice of the reviewer.
Round 2
Reviewer 2 Report
Thank you for all the corrections